# Antioxidant, Antibacterial, Enzyme Inhibitory, and Anticancer Activities and Chemical Composition of *Alpinia galanga* Flower Essential Oil

**DOI:** 10.3390/ph15091069

**Published:** 2022-08-27

**Authors:** Yufeng Tian, Xiaoyan Jia, Qinqin Wang, Tingya Lu, Guodong Deng, Minyi Tian, Ying Zhou

**Affiliations:** 1Key Laboratory of Plant Resource Conservation and Germplasm Innovation in Mountainous Region (Ministry of Education), Guizhou University, Guiyang 550025, China; 2National & Local Joint Engineering Research Center for the Exploitation of Homology Resources of Southwest Medicine and Food, Guizhou University, Guiyang 550025, China; 3College of Pharmacy, Guizhou University of Traditional Chinese Medicine, Guiyang 550025, China

**Keywords:** *Alpinia galanga* flower, farnesene, radical scavenging effects, antibacterial agent, enzyme inhibitors, cytotoxicity, apoptosis

## Abstract

*Alpinia galanga* is widely cultivated for its essential oil (EO), which has been used in cosmetics and perfumes. Previous studies of *A. galanga* focussed mostly on the rhizome but seldom on the flower. Therefore, this study was designed to identify the chemical composition of *A. galanga* flower EO and firstly estimate its antioxidant, antibacterial, enzyme inhibitory, and anticancer activities. According to the results of the gas chromatography with flame ionization or mass selective detection (GC-FID/MS) analysis, the most abundant component of the EO was farnesene (64.3%), followed by farnesyl acetate (3.6%), aceteugenol (3.2%), eugenol (3.1%), *E*-nerolidol (2.9%), decyl acetate (2.4%), octyl acetate (2.0%), sesquirosefuran (1.9%), (*E*)-*β*-farnesene (1.7%), and germacrene D (1.5%). For the bioactivities, the EO exhibited moderate DPPH and ABTS radical scavenging effects with IC_50_ values of 138.62 ± 3.07 μg/mL and 40.48 ± 0.49 μg/mL, respectively. Moreover, the EO showed strong-to-moderate antibacterial activities with various diameter of inhibition zone (DIZ) (8.79–14.32 mm), minimal inhibitory concentration (MIC) (3.13–6.25 mg/mL), and minimal bactericidal concentration (MBC) (6.25–12.50 mg/mL) values against *Staphylococcus aureus*, *Bacillus subtilis*, *Enterococcus faecalis*, *Pseudomonas aeruginosa*, *Escherichia coli*, and *Proteus vulgaris*. Interestingly, the EO possessed remarkable α-glucosidase inhibition (IC_50_ = 0.16 ± 0.03 mg/mL), which was equivalent to that of the positive control acarbose (IC_50_ = 0.15 ± 0.01 mg/mL) (*p* > 0.05). It showed moderate tyrosinase inhibition (IC_50_ = 0.62 ± 0.09 mg/mL) and weak inhibitory activity on acetylcholinesterase (AChE) (IC_50_ = 2.49 ± 0.24 mg/mL) and butyrylcholinesterase (BChE) (IC_50_ = 10.14 ± 0.59 mg/mL). Furthermore, the EO exhibited considerable selective cytotoxicity to K562 cells (IC_50_ = 41.55 ± 2.28 μg/mL) and lower cytotoxicity to non-cancerous L929 cells (IC_50_ = 120.54 ± 8.37 μg/mL), and it induced K562 cell apoptosis in a dose-dependent manner. Hence, *A. galanga* flower EO could be regarded as a bioactive natural product with great application potential in the pharmaceutical field.

## 1. Introduction

Essential oils are secondary plant metabolites with strong odors and are composed of a variety of volatile compounds [1,2]. Approximately 3000 essential oils are generated from more than 2000 plants, among which 300 are valuable from a commercial standpoint [3]. Essential oils have been employed extensively in pharmaceutical, agriculture, food, perfume, cosmetic, and sanitary industries due to their various pharmacological and biological effects, including antibacterial, antioxidant, anticancer, antidiabetic, virucidal, fungicidal, anti-inflammatory, analgesic, antimutagenic, and antiprotozoal activities [3,4,5,6]. As synthetic products may have adverse health and environmental effects, this has prompted the search for more natural and healthier alternatives, such as the utilization of essential oils [5,6,7].

*Alpinia**galanga* (L.) Willd., also called greater galangal, is a perennial herb that is cultivated primarily in Asia as an essential source of cosmetics, medicines, and culinary products [8,9,10]. The *A. galanga* rhizome has been extensively used as a spice and food-flavoring agent, as well as in Chinese, Ayurveda, Thai, and Unani traditional medicines for the treatment of various diseases, such as stomach ache, vomiting, diarrhea, diabetes, microbial infections, bronchitis, fever, headache, sore throat, whooping cough, kidney disorders, ulcer, rheumatism, and chronic enteritis [11,12,13,14,15]. The fruit of *A.*
*galanga*, which is called *Hongdoukou* in China and is listed in the Pharmacopoeia of the People’s Republic of China, has the effect of strengthening the stomach and promoting digestion and is used to treat abdominal pain, indigestion, vomiting, nausea, diarrhea, dysentery, and excessive drinking [16,17,18]. Its fragrant flower is often eaten raw or pickled and can also be used as a spice [19,20]. The essential oils and extracts from different parts of *A. galanga*, including the flower, rhizome, leaf, and fruit, have been used as cosmetic ingredients, and are cataloged in the European Commission database for information on cosmetic substances and ingredients (CosIng) and the “Catalogue of Cosmetic Raw Materials Used (2021 Edition)”, which is approved by the China National Medical Products Administration (NMPA) [21,22,23,24]. The rhizome of *A. galanga* was extensively investigated, especially its essential oil used in cosmetics and perfumes [25,26]. Its rhizome essential oil was reported as possessing multiple pharmacological and biological activities, including antibacterial, antifungal, antioxidant, antitumor, insecticidal, repellent, antifeedant, and anti-inflammatory activities [27,28,29,30,31,32,33,34,35,36].

*A. galanga* usually blooms from June to August in the subtropics, but it can bloom year-round in the tropics [8]. Its flower can be used as a vegetable, spice, and cosmetic ingredient. However, there are few studies on the *A. galanga* flower, only showing that its hexane, ethanol, and methanol extracts have antibacterial, antimicrobial, and antioxidant effects [8,11,37,38]. Nevertheless, no studies have been conducted on the bioactivities of flower essential oil, which may limit the utilization of its flower in industry. Therefore, the purpose of this research was to determine the chemical constituents of *A. galanga* flower essential oil and firstly assess its antioxidant, antibacterial, enzyme inhibitory, and anticancer activities.

## 2. Results and Discussion

### 2.1. Chemical Constituents

Relative to the fresh weight of *A. galanga* flowers, the extraction yield of the hydrodistilled EO was 0.11% (*w*/*w*). A total of fifty-seven chemical constituents were identified using GC-MS/FID analysis, which accounted for 96.0% of the total EO (Table 1). The most abundant component of the EO was farnesene (64.3%), followed by farnesyl acetate (3.6%), aceteugenol (3.2%), eugenol (3.1%), *E*-nerolidol (2.9%), decyl acetate (2.4%), octyl acetate (2.0%), sesquirosefuran (1.9%), (*E*)-*β*-farnesene (1.7%), and germacrene D (1.5%) (Figure 1). According to a previous study, the *A. galanga* flower essential oils collected from a subtropical region (Pantnagar, India) and a subtemperate region (Purara, India) were dominated by *β*-pinene (12.8 and 10.5%, resp.), 1,8-cineole (18.4 and 9.4%, resp.), *cis*-sabinene hydrate (0 and 8.3%, resp.), *α*-terpineol (4.5 and 3.4%, resp.), and (*E*)-methyl cinnamate (19.7 and 7.1%, resp.) [39]. In addition, the major volatile compounds of the hexane extract from *A. galanga* flowers (obtained from Gainesville, USA) were *α*-humulane, pentadecane, *β*-farnesol, dimethyl trisulfide, and mercaptomethylbutanol [8]. In contrast to the above-mentioned studies, the chemical composition of *A. galanga* flower EO in this study was quite different, which may have been caused by various factors, such as climatic conditions, growth conditions, developmental stages, and genetic factors.

### 2.2. Antioxidant Activity

DPPH and ABTS radical scavenging tests were used to assess the antioxidant activity of *A. galanga* flower EO, and the findings are presented in Table 2. Compared with BHT and ascorbic acid, the EO exhibited moderate DPPH and ABTS radical scavenging effects, with IC_50_ values of 138.62 ± 3.07 μg/mL and 40.48 ± 0.49 μg/mL, respectively. Eugenol, which is a phenolic compound used as a spice, has been well established as possessing potent antioxidant and free-radical-scavenging activities in both in vivo and in vitro studies [40,41]. Aceteugenol, which is an acetylated derivative of eugenol, displayed significant activity in scavenging DPPH free radicals, with an IC_50_ of 0.12 ± 0.03 μmol/L, and it showed a synergistic effect with eugenol regarding inhibiting the oxidation of sunflower oil [42]. Furthermore, isoeugenol is an isomer of eugenol, and methyleugenol is a methylated derivative of eugenol. These eugenol-related compounds may play a key role in the antioxidant activity of *A. galanga* flower EO. According to past reports, farnesyl acetate, *E*-nerolidol, octyl acetate, and germacrene D exhibit moderate-to-weak antioxidant effects [43,44,45]. Hence, these compounds could be responsible for the moderate antioxidant activity of *A. galanga* flower EO.

### 2.3. Antibacterial Activity

The antibacterial properties of *A. galanga* flower EO were tested using diameter of inhibition zone (DIZ), minimum inhibition concentration (MIC), and minimum bactericidal concentration (MBC) values (Table 3). The EO exhibited a broad-spectrum antibacterial property against the tested bacterial strains, with DIZ values ranging from 8.79 to 14.32 mm. Based on previous research, MIC values under 5 mg/mL are thought to have strong antibacterial activity [46]. Hence, the EO displayed a strong antibacterial effect against *Staphylococcus aureus* (MIC = 3.13 mg/mL, MBC = 6.25 mg/mL), *Bacillus subtilis* (MIC = 3.13 mg/mL, MBC = 6.25 mg/mL), *Pseudomonas aeruginosa* (MIC = 3.13 mg/mL, MBC = 12.50 mg/mL), and *Proteus vulgaris* (MIC = 3.13 mg/mL, MBC = 6.25 mg/mL), and showed moderate antibacterial property against *Enterococcus faecalis* (MIC = 6.25 mg/mL, MBC = 12.50 mg/mL) and *Escherichia coli* (MIC = 6.25 mg/mL, MBC = 12.50 mg/mL). Several studies attributed the antibacterial activity of essential oils to the active ingredient farnesene [47,48]. In a previous study, farnesyl acetate showed significant antibacterial activity against *Acinetobacter baumannii*, *Klebsiella pneumoniae*, *Escherichia coli*, *Staphylococcus aureus*, and *Enterococcus faecalis*, with MIC values ranging from 20 µg/mL to 28 µg/mL [49]. Additionally, other major constituents, including aceteugenol, eugenol, *E*-nerolidol, and germacrene D, were shown to possess antibacterial properties [45,50,51,52]. Therefore, the significant antibacterial activity of the EO can be explained by the presence of these main components. According to these findings, *A. galanga* flower EO could be employed in the pharmaceutical field as a novel natural antibacterial agent.

### 2.4. Enzyme Inhibitory Activity

The *A. galanga* flower EO was tested for its ability to inhibit the following enzymes: α-glucosidase, tyrosinase, acetylcholinesterase (AChE), and butyrylcholinesterase (BChE). All results are summarized in Table 4.

The *A. galanga* flower EO exhibited remarkable α-glucosidase inhibitory ability (IC_50_ = 0.16 ± 0.03 mg/mL), and its inhibitory effect was equivalent to that of the positive control acarbose (IC_50_ = 0.15 ± 0.01 mg/mL) (*p* > 0.05). Acarbose, which is an α-glucosidase inhibitor, is widely used to treat type 2 diabetes, as it can lower postprandial insulin and blood glucose levels by delaying carbohydrate absorption [53]. Concerns about acarbose’s side effects, such as flatulence, diarrhea, and abdominal distension, have prompted the search for natural products that inhibit α-glucosidase’s activity as complementary/alternative treatments for type 2 diabetes [54]. An earlier study demonstrated that farnesene, which is the most predominant component of the *A. galanga* flower EO, exerted significant *α*-glycosidase inhibitory activity [55]. Farnesyl acetate reduced postprandial blood glucose levels and was a potential α-glucosidase inhibitor [56]. In addition, past studies demonstrated that other major components, including aceteugenol, eugenol, and *E*-nerolidol, had *α*-glycosidase inhibitory effects [57,58,59]. Thus, the remarkable α-glucosidase inhibitory activity of *A. galanga* flower EO could be attributed to these main constituents, and it could be used in the pharmaceutical industry as a new source of natural α-glucosidase inhibitors.

As depicted in Table 4, compared with the positive control arbutin (IC_50_ = 0.19 ± 0.06 mg/mL), *A. galanga* flower EO showed a moderate tyrosinase inhibitory effect (IC_50_ = 0.62 ± 0.09 mg/mL). Tyrosinase is a crucial enzyme in the enzymatic browning of fruits and mammalian melanogenesis [5]. Eugenol had a significant inhibitory effect on tyrosinase and could be used as a tyrosinase inhibitor [60]. Moreover, according to the research of Arung et al., eugenol and eugenol acetate significantly inhibit melanin formation in B16 melanoma cells [61]. Hence, the moderate tyrosinase inhibition of *A. galanga* flower EO may have been due to these main components.

As shown in Table 4, compared with the positive reference galanthamine, *A. galanga* flower EO showed weak inhibitory activity against AChE and BChE, with IC_50_ values of 2.49 ± 0.24 mg/mL and 10.14 ± 0.59 mg/mL, respectively. Cholinesterase inhibitors enhance cholinergic neurotransmission by inhibiting the decomposition of acetylcholine, which has become an effective strategy for the treatment of Alzheimer’s disease [62]. Eugenol, which is a major component of EO, was reported to display significant inhibition of AChE (IC_50_: 42.44 ± 1.21 µg/mL) and BChE (IC_50_: 63.51 ± 1.88 µg/mL) [63]. *E*-Nerolidol significantly reduces the AChE activity and oxidative/nitrosative stress, improves locomotor activity, and reverses motor incoordination and cognitive impairment after weight-drop-induced traumatic brain injury (TBI) in rats [64]. Thus, these major compounds could be responsible for the anti-cholinesterase activity of *A. galanga* flower EO.

### 2.5. Anticancer Activity

The cytotoxic properties of *A. galanga* flower EO were investigated against human tumor cell lines (lung adenocarcinoma A549 cells, prostatic carcinoma PC-3 cells, leukemic K562 cells, and non-small-cell lung cancer NCI-H1299 cells) and a non-cancerous cell line (murine fibroblast L929 cells) using MTT assays. Cisplatin was used as a positive control. As shown in Table 5, *A. galanga* flower EO showed significant cytotoxic activities against the four human tumor cells, with IC_50_ values ranging from 41.55 ± 2.28 μg/mL to 127.37 ± 4.15 μg/mL. In particular, EO exhibited considerable selective cytotoxicity against K562 cells (IC_50_ = 41.55 ± 2.28 μg/mL), and its toxicity was almost 3 times that of non-cancerous L929 cells (IC_50_ = 120.54 ± 8.37 μg/mL). Hence, the K562 cells were selected for subsequent studies.

Cancer cells are capable of evading apoptosis; therefore, triggering apoptosis is a key strategy for cancer therapies [65]. As shown in Figure 2A, the results of phase-contrast microscopy revealed typical apoptotic morphological alterations in EO-treated K562 cells, including cell shrinkage and fragmentation. Moreover, AO/EB staining and Hoechst 33,258 staining were used to observe the changes in the nuclear morphology of the K562 cells. According to the AO/EB staining results, after the EO treatment, the percentage of apoptotic cells with orange-red fluorescent nuclei increased, while the percentage of living cells with bright green fluorescent nuclei reduced (Figure 2B). In the Hoechst 33,258 staining analysis, the nuclei of K562 cells treated with EO displayed brighter blue fluorescence, indicating that the nuclear chromatin of K562 cells was condensed with apoptotic features (Figure 2C). In addition, the Annexin V-PE/7-AAD staining analysis was performed using a flow cytometer to quantitatively estimate the EO-induced apoptosis. As shown in Figure 3A,B, the percentage of apoptotic cells (Q2 + Q3) increased dose-dependently from 6.88 ± 0.07% in the control to 9.38 ± 1.12%, 18.36 ± 0.20%, 26.47 ± 0.30%, 32.13 ± 0.50%, and 55.04 ± 0.54% at the doses of 20 μg/mL, 40 μg/mL, 60 μg/mL, 80 μg/mL, and 160 μg/mL of EO, respectively. These results suggested that EO significantly induced K562 cell apoptosis in a concentration-dependent manner.

Numerous farnesene-rich essential oils, such as those from *Zornia brasiliensis*, *Malus domestica*, and *Streblus asper*, were shown to have anticancer properties [66,67,68,69]. Farnesyl acetate had higher cytotoxicity against the tumor cells malignant melanoma MEWO (IC_50_ = 734 µM) and promyelocytic leukemia HL-60 (IC_50_ = 121 µM) compared with non-cancerous cells (fibroblasts HFIG, keratinocytes HaCaT, and epithelium of the small intestine IEC6) (IC_50_ > 1000 μM) [49]. The anticancer effects of eugenol have been well recognized, and it can induce apoptosis in different cancer cells [70]. Furthermore, the anticancer activities of other main components in EO, such as *E*-nerolidol, (*E*)-*β*-farnesene, and germacrene D, were demonstrated in previous research [71,72,73]. Thus, *A. galanga* flower EO’s significant anticancer properties could be explained by these main components. Based on these findings, *A. galanga* flower EO may serve as a new source of natural anticancer agents in the pharmaceutical field.

## 3. Materials and Methods

### 3.1. Plant Material

*Alpinia galanga* flower was harvested from Yulin City, Guangxi Province, China, in June 2020. It was identified by Prof. Guoxiong Hu from the College of Life Sciences, Guizhou University. A voucher specimen (voucher no: AG20200621) was deposited in the National and Local Joint Engineering Research Center for the Exploitation of Homology Resources of Southwest Medicine and Food, Guizhou University.

### 3.2. Preparation of Essential Oil

The fresh, finely chopped flower of *A. galanga* (2.5 kg) was hydrodistilled for 4 h in an all-glass Clevenger-type apparatus. After dehydration with anhydrous Na_2_SO_4_, the essential oil (2.76 g, 0.11% *w*/*w*) was kept in a sealed vial at 4 °C for further analysis.

### 3.3. Analysis of Essential Oil

Quantitative analysis of the EO was carried out using a gas chromatograph (GC) equipped with a flame ionization detector (FID) (model 6890, Agilent Technologies, Santa Clara, CA, USA). Column: HP-5MS capillary column (60 m × 0.25 mm, 0.25 μm film thickness). The GC settings were as follows: injection volume (1 μL), split ratio (1:20), carrier gas helium (1 mL/min), and oven temperature program (kept at 70 °C for 2 min, increased to 180 °C at 2 °C/min, raised to 310 °C at 10 °C/min, and finally kept at 310 °C for 14 min). The gas chromatograph–mass spectrometer (GC-MS) (model 6890/5975C, Agilent Technologies, Santa Clara, CA, USA) was used for qualitative analysis of the EO. GC parameters and the column were the same as in the GC-FID analysis. The MS was operated as follows: ion source temperature at 230 °C, interface temperature at 280 °C, ionization voltage at 70 eV, and a scan range of *m*/*z* 29 to 500 amu. The relative abundance (%) of the chemical constituents was determined using the peak area. Standard n-alkanes (C_9_–C_30_) were used for the calculation of the retention index (RI). Each component of EO was determined by comparing the RI and mass spectra in the Wiley 275 (Wiley, New York, NY, USA) and NIST 2020 (National Institute of Standards and Technology, Gaithersburg, MD, USA) databases.

### 3.4. Antioxidant Activity

#### 3.4.1. DPPH Assay

The 1,1-diphenyl-2-picrylhydrazyl (DPPH) radical scavenging effect was assayed by utilizing Tian et al.’s method [74]. Briefly, the DPPH solution (2 mL, 0.08 mM) was blended with the sample solution (2 mL) and then kept at room temperature for 30 min in the dark. The optical density at 517 nm was recorded. Ascorbic acid and butylated hydroxytoluene (BHT) served as positive controls. Data were presented as IC_50_ values.

#### 3.4.2. ABTS Assay

The 2,2′-azino-bis-3-ethylbenzthiazoline-6-sulphonic acid (ABTS) radical scavenging ability was determined by utilizing the method reported by Tian et al. [74]. ABTS•^+^ solution was generated by mixing ABTS solution (50 mL, 0.7 mM) with K_2_S_2_O_8_ solution (50 mL, 2.45 mM) and incubating at room temperature for 12 h in the dark. Subsequently, methanol was used to dilute the ABTS•^+^ solution, yielding an absorbance of 0.70 ± 0.02 at 734 nm. The sample solution (0.4 mL) and diluted ABTS•^+^ solution (4 mL) were blended and incubated at room temperature for 10 min in the dark, and then the optical density at 734 nm was measured. BHT and ascorbic acid were used as positive controls. The results were expressed as IC_50_ values.

### 3.5. Antibacterial Activity

#### 3.5.1. Bacterial Strains

The EO’s antibacterial effect was evaluated against the following bacterial strains: *Staphylococcus aureus* (ATCC 6538P), *Bacillus subtilis* (ATCC 6633), *Enterococcus faecalis* (ATCC 19433), *Pseudomonas aeruginosa* (ATCC 9027), *Escherichia coli* (CICC 10389), and *Proteus vulgaris* (ACCC 11002).

#### 3.5.2. Disc Diffusion Assay

The disc diffusion method was employed to determine the diameter of inhibition zone (DIZ) [5]. In brief, bacterial suspension (100 μL, 1 × 10^6^ CFU/mL) was evenly spread on Mueller–Hinton agar plates. Afterward, the filter paper discs (diameter 6 mm) with 20 μL of pure EO or streptomycin (100 μg/mL) were added and incubated for 24 h at 37 °C. Finally, the DIZ was measured and recorded.

#### 3.5.3. MIC and MBC Assays

The microdilution broth assay was utilized to detect the minimum inhibitory concentration (MIC) and minimum bactericidal concentration (MBC) values [5]. The EO solution (100 mg/mL, *w*/*v* in 0.1% DMSO) was two-fold serially diluted with the medium to concentrations of 50.00, 25.00, 12.50, 6.25, 3.13, 1.56, 0.78, 0.39, 0.20, and 0.10 mg/mL. Streptomycin solution (100 μg/mL, *w*/*v* in distilled water) was also diluted with the medium to the dose range of 0.10–50 μg/mL. The bacterial suspensions (100 μL) were seeded into each well of 96-well plates at a density of 5 × 10^4^ CFU/well. Subsequently, 100 μL of the diluted EO or streptomycin solution was added and cultured at 37 °C for 24 h. Resazurin aqueous solution (10 μL, 0.01%) was utilized as a microbial growth indicator, added to each well, and incubated in the dark for 2 h at 37 °C. The minimum sample concentration that did not cause a color change was its MIC value. Furthermore, 10 μL of culture from the wells with no change in color was spread on the Mueller–Hinton agar plate and inoculated at 37 °C for 24 h. The MBC values were determined as the lowest concentration of EO that induced no visible growth of the tested bacteria.

### 3.6. Enzyme Inhibitory Activities

#### 3.6.1. α-Glucosidase Inhibitory Activity

The α-glucosidase inhibitory effect was performed following the protocol reported by Hong et al. [5]. A total of 90 μL of EO solution or acarbose solution (positive control) was mixed with α-glucosidase solution (10 μL, 0.8 U/mL) and added to each well of the 96-well plates. After incubation at 37 °C for 15 min, the reaction was started by adding 10 μL of *p*-nitrophenyl-*α*-D-glucopyranoside (p-NPG) substrate (1 mM) and maintained at 37 °C for 15 min, then halted by the addition of 80 μL Na_2_CO_3_ solution (0.2 M). Subsequently, the optical density at 734 nm was recorded using a microplate reader (Varioskan Lux Multimode, Thermo Fisher Scientific, Waltham, MA, USA). The results of the α-glucosidase inhibitory activity were presented using IC_50_ values.

#### 3.6.2. Tyrosinase Inhibitory Activity

The tyrosinase inhibition was performed based on the protocol reported by Tian et al. and utilized arbutin as a positive control [74]. A total of 70 μL of EO solution and 100 μL of tyrosinase solution (100 U/mL) was mixed and incubated at 37 °C for 5 min in a 96-well plate. After that, the L-tyrosine substrate (80 L, 5.5 mM) was added to start the reaction and incubated at 37 °C for 30 min. Then, the optical density at 492 nm was recorded, and the tyrosinase inhibitory activity was presented using IC_50_ values.

#### 3.6.3. Cholinesterase Inhibitory Activity

The acetylcholinesterase (AChE) and butyrylcholinesterase (BChE) inhibitory effects were assessed based on Ellman’s method with a slight modification [75]. An AChE or BuChE solution (10 μL, 0.5 U/mL, pH 8.0) and EO solution (50 μL) were blended and maintained at 4 °C for 15 min in a 96-well plate. Then, 5,5′-dithiobis-(2-nitrobenzoic acid) (DTNB) solution (20 μL, 2 mM, pH 8.0) and acetylthiocholine (ATCI) or butyrylthiocholine (BTCI) solution (20 μL, 2 mM) were added to initiate the reaction. After incubating at 37 °C for 30 min, the absorbance at 405 nm was recorded. The galanthamine was used as a positive reference. The AChE and BChE inhibitory activities were presented using IC_50_ values.

### 3.7. Anticancer Activity

#### 3.7.1. Cytotoxic Activity

The cytotoxic activities against human cancerous cells (A549 lung adenocarcinoma cells, PC-3 prostatic carcinoma cells, K562 leukemic cells, and NCI-H1299 non-small-cell lung cancer cells) and non-cancerous cells (L929 murine fibroblast cells) were evaluated using MTT assays with a slight modification [76]. The EO solution (160 mg/mL, *w*/*v* in DMSO) or cisplatin solution (40 mg/mL, *w*/*v* in DMSO) was diluted with RPMI-1640 medium, where the final concentration of DMSO was lower than 0.1%. The cell suspensions (80 μL) were seeded into a 96-well plate (5 × 10^3^ cells/well). After 24 h of incubation, cells were treated with EO (0, 10, 20, 40, 80, and 160 µg/mL) or cisplatin (0, 1.88, 3.75, 7.5, 15, and 30 µg/mL) for 48 h. Subsequently, MTT solutions (10 μL, 5 mg/mL in PBS) were added and incubated for 4 h. Finally, the formazan crystals in each well were dissolved in 150 μL of DMSO, and a microplate reader was used to detect the absorbance at 490 nm. The results of the cytotoxicity were presented as IC_50_ values.

#### 3.7.2. Morphology Assay

K562 cells were seeded into a 6-well plate (5 × 10^5^ cells/well) and cultured for 24 h. Subsequently, the cells were subjected to the new medium with different doses of EO (0, 20, 40, 80, and 160 µg/mL) for 48 h. Finally, morphological alterations of K562 cells were recorded using a Leica DMi8 inverted microscope (Leica Microsystems, Germany).

Acridine orange/ethidium bromide (AO/EB) staining and Hoechst 33,258 staining assays were used to detect the nuclear morphology changes of the K562 cells. After the treatment with EO as mentioned above, the cell suspensions (1.5 mL) were collected, centrifuged at 1000× g for 5 min, fixed with 4% paraformaldehyde (0.5 mL) for 10 min, washed twice with PBS (1 mL), and resuspended in 50 μL PBS. Afterward, the cell suspension (10 μL) was dropped onto the glass slides. For AO/EB staining, cells were stained with AO/EB dye (10 µg/mL AO and 10 µg/mL EB) for 5 min. After washing twice with PBS (0.5 mL), the stained cells were viewed using a fluorescence microscope. For Hoechst 33,258 staining, K562 cells were stained with 500 μL of Hoechst 33,258 for 5 min, washed twice with PBS (0.5 mL), and recorded using a Leica DMi8 inverted fluorescence microscope (Leica Microsystems, Germany).

#### 3.7.3. Flow Cytometry Assay

Quantitative analysis of the EO-induced apoptosis of K562 cells was carried out using the Annexin V-PE/7-AAD apoptosis kit (Multi Sciences (Lianke) Biotech, Co., Ltd., Hangzhou, China) based on the manufacturer’s instructions. Briefly, cells were seeded into a 6-well plate (3 × 10^5^ cells/well), incubated for 24 h, and then treated with a new medium supplemented with EO at different doses of 0, 20, 40, 60, 80, and 160 µg/mL. After 48 h of incubation, cells were collected, rinsed with pre-cold PBS, and resuspended in 1× binding buffer (500 µL). Then, the cells were labeled with Annexin V-PE (5 µL) and 7-AAD (10 µL) for 5 min in the dark at room temperature. The apoptosis rate of cells was analyzed using an ACEA NovoCyte^TM^ flow cytometer (ACEA Biosciences, San Diego, CA, USA).

### 3.8. Statistical Analysis

Each experiment was repeated at least three times independently, and the results were presented as the mean ± SD (standard deviation). SPSS software (version 25.0) was used for the statistical analysis. The significant differences between groups were analyzed using one-way analysis of variance (ANOVA) and the least significant difference (LSD) test at * *p* < 0.05, ** *p* < 0.01, and *** *p* < 0.001.

## 4. Conclusions

To our knowledge, this is the first study on the antioxidant, antibacterial, enzyme inhibitory, and anticancer activities of *A. galanga* flower essential oil. Fifty-seven chemical components were identified using GC-FID/MS. For the bioactivities, the *A. galanga* flower EO exhibited moderate DPPH and ABTS radical scavenging effects. Moreover, it showed strong-to-moderate antibacterial activities against *S. aureus*, *B. subtilis*, *E. faecalis*, *P. aeruginosa*, *E. coli*, and *P. vulgaris*. In addition to the weak inhibition against AChE and BChE, as well as moderate tyrosinase inhibition, an interesting finding was that it displayed remarkable α-glucosidase inhibition comparable to that of acarbose. Furthermore, it exhibited considerable selective cytotoxicity to K562 cells and lower cytotoxicity to non-cancerous L929 cells, and it induced K562 cell apoptosis in a concentration-dependent manner. Hence, *A. galanga* flower EO could be regarded as a bioactive natural product with great application potential in the pharmaceutical field.

## Figures and Tables

**Figure 1 pharmaceuticals-15-01069-f001:**
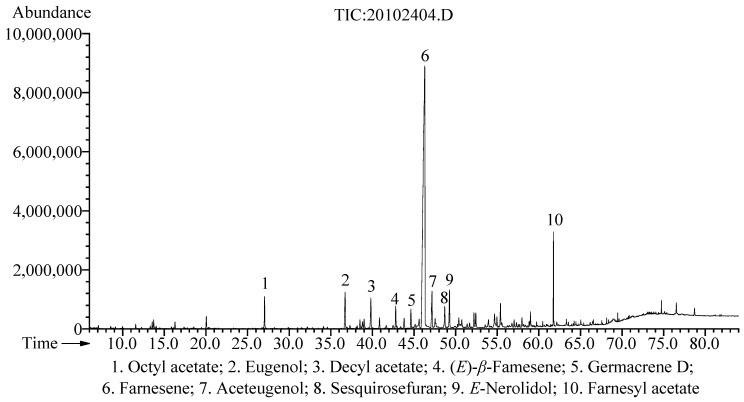
GC-MS chromatogram of the essential oil from *A. galanga* flowers.

**Figure 2 pharmaceuticals-15-01069-f002:**
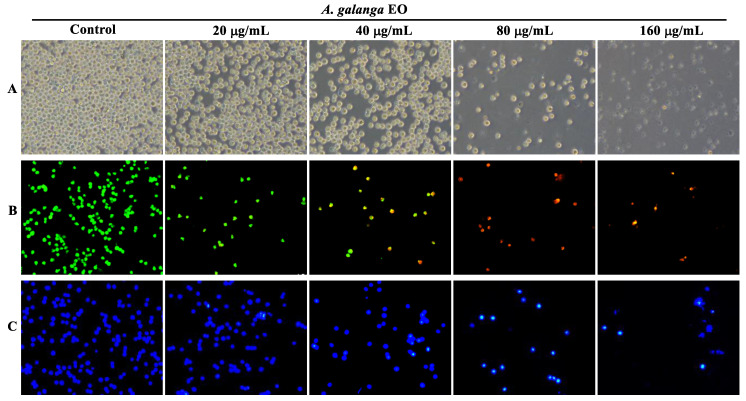
Effect of *A. galanga* flower essential oil on the morphological changes of K562 cells. (**A**) A phase-contrast microscope was used to observe the morphological alterations of K562 cells (100× magnification). AO/EB staining (**B**) and Hoechst 33,258 staining (**C**) were employed to assess the nuclear morphological alterations of K562 cells, which were viewed under a fluorescence microscope with 100× magnification.

**Figure 3 pharmaceuticals-15-01069-f003:**
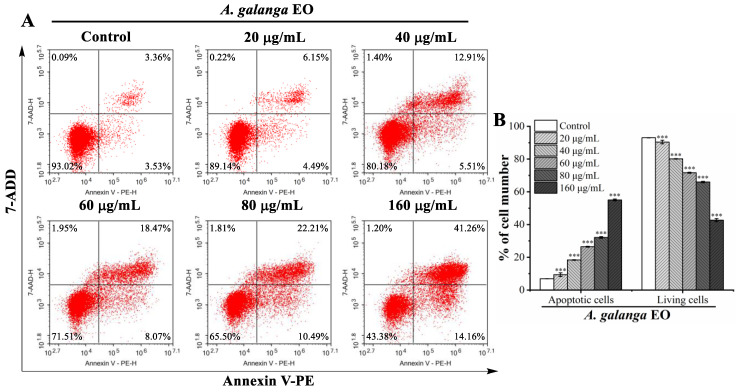
Flow cytometry apoptosis analysis of K562 cells treated with *A. galanga* flower essential oil. (**A**) K562 cells were labeled with Annexin V-PE and 7-AAD and then detected using flow cytometry. Cells in the upper-left quadrant (Q1-UL: Annexin V-PE–/7-AAD+): necrotic cells; upper-right quadrant (Q2-UR: Annexin V-PE+/7-AAD+): late apoptotic cells; lower-right quadrant (Q3-LR: Annexin V-PE+/7-AAD–): early apoptotic cells; lower-left quadrant (Q4-LL: Annexin V-PE–/7-AAD–): living cells. (**B**) Percentage of apoptotic cells and living cells. Results are presented as the mean ± SD. *** *p* < 0.001 versus the control group.

**Table 1 pharmaceuticals-15-01069-t001:** Chemical compounds of the essential oil from *A. galanga* flowers.

Compound ^a^	RT (min)	RI ^b^	RI ^c^	%Area	Identification ^d^
Octane	7.066	800	800	0.1	RI, MS
Ethylbenzene	8.877	855	862	tr ^e^	RI, MS
*p*-Xylene	9.122	865	871	0.1	RI, MS
Nonane	9.961	900	900	0.1	RI, MS
*α*-Pinene	11.56	937	936	0.2	RI, MS
Camphene	12.257	952	952	0.1	RI, MS
Sabinen	13.306	974	976	0.2	RI, MS
*β*-Pinene	13.545	979	981	0.4	RI, MS
Sulcatone	13.714	986	985	0.6	RI, MS
2,2,4,6,6-Pentamethylheptane	14.02	990	992	0.2	RI, MS
Decane	14.356	1000	1000	tr ^e^	RI, MS
*α*-Phellandrene	14.831	1005	1008	tr ^e^	RI, MS
*p*-Cymene	15.86	1025	1026	0.1	RI, MS
*D*-Limonene	16.113	1031	1031	tr ^e^	RI, MS
Eucalyptol	16.281	1032	1034	0.4	RI, MS
Melonal	17.365	1054	1053	0.1	RI, MS
*γ*-Terpinene	17.768	1060	1060	tr ^e^	RI, MS
1-Octanol	18.214	1070	1068	tr ^e^	RI, MS
3-Methylbenzaldehyde	18.399	1071	1071	tr ^e^	RI, MS
Linalool oxide	18.543	1074	1074	0.1	RI, MS
Ethyl 2-(5-methyl-5-vinyltetrahydrofuran-2-yl)propan-2-yl carbonate	19.47	1090	1090	tr ^e^	RI, MS
Linalool	20.049	1099	1100	0.7	RI, MS
6-Methyl-3,5-heptadiene-2-one	20.34	1107	1105	0.1	RI, MS
Terpinen-4-ol	25.136	1177	1180	tr ^e^	RI, MS
*α*-Terpineol	25.959	1189	1193	tr ^e^	RI, MS
Decanal	26.725	1206	1205	0.1	RI, MS
*cis*-5-Octenyl acetate	26.88	1206	1208	0.1	RI, MS
Octyl acetate	27.044	1210	1211	2.0	RI, MS
Geraniol	29.888	1255	1255	0.1	RI, MS
*α*-Citral	30.993	1270	1272	0.1	RI, MS
Nonanol acetate	33.492	1309	1311	tr ^e^	RI, MS
Eugenol	36.71	1358	1361	3.1	RI, MS
Cerulignol	37.301	1373	1370	0.2	RI, MS
Copaene	38.038	1376	1382	0.1	RI, MS
Geranyl acetate	38.165	1382	1384	0.2	RI, MS
*β*-Elemen	39.009	1391	1397	0.6	RI, MS
Methyleugenol	39.546	1403	1406	0.1	RI, MS
Decyl acetate	39.815	1409	1410	2.4	RI, MS
Caryophyllene	40.853	1419	1427	0.8	RI, MS
*trans*-Bergamotene	41.662	1435	1440	0.2	RI, MS
Isoeugenol	42.473	1450	1453	0.3	RI, MS
(*E*)-*β*-Farnesene	42.804	1457	1459	1.7	RI, MS
Humulene	42.949	1460	1461	0.2	RI, MS
*epi*-*β*-Caryophyllene	43.404	1466	1469	0.1	RI, MS
Germacrene D	44.631	1481	1489	1.5	RI, MS
(*Z*)-*α*-Farnesene	45.135	1491	1497	0.3	RI, MS
Farnesene	46.296	1508	1516	64.3	RI, MS
Aceteugenol	47.163	1524	1531	3.2	RI, MS
Sesquirosefuran	48.691	1557	1557	1.9	RI, MS
*E*-Nerolidol	49.266	1564	1567	2.9	RI, MS
Germacrene D-4-ol	50.233	1574	1584	0.2	RI, MS
Spathulenol	50.387	1576	1586	0.5	RI, MS
Caryophyllene oxide	50.74	1581	1592	0.4	RI, MS
*α*-Cadinol	54.673	1653	1663	0.8	RI, MS
*trans*-Farnesal	58.842	1745	1754	0.5	RI, MS
*cis*-9-Hexadecenal	60.467	1803	1796	0.2	RI, MS
Farnesyl acetate	61.754	1843	1848	3.6	RI, MS
Monoterpene hydrocarbons				1.08	
Oxygenated monoterpenes				1.59	
Sesquiterpene hydrocarbons				69.64	
Oxygenated sesquiterpenes				10.75	
Others				12.97	
Total (%)				96.0	
Yield (*w*/*w*) (%)				0.11	

^a^ Compounds listed based on their elution order on an HP-5MS column. ^b^ Retention index (RI) from NIST 2020 mass spectral databases. ^c^ RI calculated using C_8_–C_30_ n-alkanes. ^d^ Identification: RI, matching calculated RI to the RI in the Wiley 275 and NIST 2020 databases; MS, based on a comparison with the Wiley 275 and NIST 2020 MS databases. ^e^ tr: trace (trace < 0.1%).

**Table 2 pharmaceuticals-15-01069-t002:** Antioxidant activity of *A. galanga* flower essential oil.

Samples	Antioxidant Activity (IC_50_, μg/mL) ^1^	
	DPPH	ABTS
Essential oil	138.62 ± 3.07 ^a^	40.48 ± 0.49 ^a^
BHT ^2^	14.16 ± 0.30 ^b^	1.99 ± 0.05 ^b^
Ascorbic acid ^2^	0.52 ± 0.01 ^c^	1.05 ± 0.02 ^c^

^1^ IC_50_: the sample concentration required to achieve a 50% free-radical-scavenging efficiency in the test. ^2^ BHT and ascorbic acid served as positive controls. ^a–c^ A significant difference (*p* < 0.05) is indicated by different letters in the same column.

**Table 3 pharmaceuticals-15-01069-t003:** Antibacterial activity of *A. galanga flower* essential oil.

Bacterial Strains ^a^	Essential Oil			Streptomycin		
	DIZ ^b^ (mm)	MIC ^c^ (mg/mL)	MBC ^c^ (mg/mL)	DIZ ^b^ (mm)	MIC ^c^ (μg/mL)	MBC ^c^ (μg/mL)
**Gram-positive**						
*S. aureus*	10.98 ± 1.14	3.13	6.25	19.78 ± 0.29	0.78	1.56
*B. subtilis*	14.32 ± 2.81	3.13	6.25	18.43 ± 0.82	0.78	1.56
*E. faecalis*	9.21 ± 0.92	6.25	12.50	8.38 ± 0.34	12.50	25.00
**Gram-negative**						
*P. aeruginosa*	9.54 ± 0.20	3.13	12.50	10.35 ± 0.19	1.56	3.13
*E. coli*	8.79 ± 0.49	6.25	12.50	11.83 ± 0.40	1.56	6.25
*P. vulgaris*	9.57 ± 0.42	3.13	6.25	16.92 ± 0.54	0.78	6.25

^a^ Bacterial strains: *Staphylococcus aureus* (ATCC 6538P), *Bacillus subtilis* (ATCC 6633), *Enterococcus faecalis* (ATCC 19433), *Pseudomonas aeruginosa* (ATCC 9027), *Escherichia coli* (CICC 10389), and *Proteus vulgaris* (ACCC 11002). ^b^ DIZ: diameter of inhibition zone (mm, including 6 mm disk). Disks contained 20 µL of pure EO or streptomycin (100 µg/mL, *w*/*v* in distilled water). ^c^ MIC: minimal inhibitory concentration; MBC: minimal bactericidal concentration.

**Table 4 pharmaceuticals-15-01069-t004:** The enzyme inhibitory activity of *A. galanga* flower essential oil.

Samples	Enzyme Inhibitory Effect (IC_50_, mg/mL) ^1^			
	α-Glucosidase	Tyrosinase	Acetylcholinesterase	Butyrylcholinesterase
Essential oil	0.16 ± 0.03 ^a^	0.62 ± 0.09 ^a^	2.49 ± 0.24 ^a^	10.14 ± 0.59 ^a^
Acarbose	0.15 ± 0.01 ^a^	–	–	–
Arbutin	–	0.19 ± 0.06 ^b^	–	–
Galanthamine *	–	–	0.25 ± 0.06 ^b^	4.65 ± 0.16 ^b^

^1^ IC_50_: the dose of the sample that inhibits 50% of enzyme activities. ^a,b^ Significant differences (*p* < 0.05) are indicated by different letters in the same column. * Galanthamine: IC_50_ (μg/mL).

**Table 5 pharmaceuticals-15-01069-t005:** Cytotoxic activity of *A. galanga* flower essential oil.

Samples	Cell Line (IC_50_, µg/mL) ^1^				
	A549	PC-3	K562	NCI-H1299	L929
Essential oil	102.09 ± 3.86 ^a^	97.09 ± 5.02 ^a^	41.55 ± 2.28 ^b^	127.37 ± 4.15 ^c^	120.54 ± 8.37 ^c^
Cisplatin	15.13 ± 0.72 ^a^	10.69 ± 0.69 ^b^	6.32 ± 0.77 ^c^	4.78 ± 0.93 ^d^	9.16 ± 0.64 ^e^

^1^ IC_50_: the concentration of sample that reduces the cell growth by 50%. Cisplatin was used as a positive control. Cell lines: A549 (human lung adenocarcinoma cells), PC-3 (human prostatic carcinoma cells), K562 (human leukemic cells), NCI-H1299 (human non-small cell lung cancer cells), and L929 (murine fibroblast cells). ^a–e^ Significant differences (*p* < 0.05) are indicated by different letters in the same row.

## Data Availability

Data is contained within the article.

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
