# Peer review of "Antioxidant, Antibacterial, Enzyme Inhibitory, and Anticancer Activities and Chemical Composition of Alpinia galanga Flower Essential Oil"

_pharmaceuticals, 2022, doi:10.3390/ph15091069_

Round 1

Reviewer 1 Report

The manuscript presents interesting results of a study conducted on an essential oil derived from Alpinia galanga flowers. The essential oil was obtained by hydrodistillation of the flowers. Its composition was found to be markedly different from that of EO from analogous flowers, but growing in a different region. This represents a very important observation, especially with regard to the use of plants or their parts as potential pharmaceuticals. A number of studies have also been conducted to determine the biological activities of the resulting oil. The results of some studies are very promising.

The manuscript is interesting and well-written, but needs some revisions.

Minor remarks

Table 1 - the main components of EO could be labeled, for example, in bold, they would then be more visible.

Section 2.5, line 194 - only the cell line designations are included here, there is no information on what type of cancer these designations represent. This should be supplemented.

Section 3.5.3 and section 3.7.1 - it is mentioned here that a series of dilutions were performed, but it is not specifically stated which ones. The missing information should be completed.

Author Response

Chemical Composition, Antioxidant, Antibacterial, Enzyme Inhibitory, and Anticancer Activities of Alpinia galanga Flower Essential Oil

(Manuscript ID: pharmaceuticals-1886802)

Dear Reviewer:

  Thanks for your comments concerning our manuscript entitled “Chemical Composition, Antioxidant, Antibacterial, Enzyme Inhibitory, and Anticancer Activities of Alpinia galanga Flower Essential Oil” (Manuscript ID: pharmaceuticals-1886802). Those comments are all valuable and very helpful for revising and improving our paper, as well as the important guiding significance to our researches. We have studied comments carefully and have made correction which we hope meet with approval. Revised portion are marked in red in the paper. The main corrections in the paper and the responds to the reviewer’s comments are as flowing:

Responds to the reviewer’s comments:

For the first reviewer:

Reviewer 1: Comments to the Author

Comments:

The manuscript presents interesting results of a study conducted on an essential oil derived from Alpinia galangal flowers. The essential oil was obtained by hydrodistillation of the flowers. Its composition was found to be markedly different from that of EO from analogous flowers, but growing in a different region. This represents a very important observation, especially with regard to the use of plants or their parts as potential pharmaceuticals. A number of studies have also been conducted to determine the biological activities of the resulting oil. The results of some studies are very promising.

Reply: Thank you very much for the positive comments on our work.

Suggestion 1: “Table 1 - the main components of EO could be labeled, for example, in bold, they would then be more visible.

Response: We have bolded the main components of EO in Table 1. Revised portions are marked in red in the manuscript. Thank you for these positive and constructive comments and suggestions.

Suggestion 2: “Section 2.5, line 194 - only the cell line designations are included here, there is no information on what type of cancer these designations represent. This should be supplemented.

Response: “human tumor cell lines (A549, PC-3, K562, and NCI-H1299) and non-cancerous cell line (L929)” has been replaced by “human tumor cell lines (lung adenocarcinoma A549 cells, prostatic carcinoma PC-3 cells, leukemic K562 cells, and non-small cell lung cancer NCI-H1299 cells) and non-cancerous cell line (murine fibroblast L929 cells)”. Revised portions are marked in red in the paper. Thank you for these positive and constructive comments and suggestions.

Suggestion 3: “Section 3.5.3 and section 3.7.1 - it is mentioned here that a series of dilutions were performed, but it is not specifically stated which ones. The missing information should be completed”.

Response: In the section 3.5.3, “EO solution (0.1 % DMSO) was two-fold serially diluted with the medium.” has been replaced by “EO solution (100 mg/mL, w/v in 0.1 % DMSO) was two-fold serially diluted with the medium to concentrations of 50.00, 25.00, 12.50, 6.25, 3.13, 1.56, 0.78, 0.39, 0.20, and 0.10 mg/mL. Streptomycin solution (100 μg/mL, w/v in distilled water) was also diluted with medium to the dose range 0.10–50 μg/mL.”. Additionally, in line 315, “diluted EO solution (100 μL) was added and incubated for 24 h at 37 °C” has been replaced by “100 μL of diluted EO solution or streptomycin solution was added and incubated for 24 h at 37 °C”.

In the section 3.7.1, “EO dissolved in DMSO was two-fold serially diluted with RPMI-1640 medium, where the final concentration of DMSO was lower than 0.05%, and cisplatin was used as a positive control.” has been replaced by “EO solution (160 mg/mL, w/v in DMSO) or cisplatin solution (40 mg/mL, w/v in DMSO) was diluted with RPMI-1640 medium, where the final concentration of DMSO was lower than 0.1%.”. Additionally, “After 24 h of incubation, the diluted EO solution (20 mL) was added to each well and cultured for 48 h.” has been replaced by “After 24 h of incubation, cells were treated with EO (0, 10, 20, 40, 80, and 160 µg/mL) or cisplatin (0, 1.88, 3.75, 7.5, 15, and 30 µg/mL) for 48 h.”. Revised portions are marked in red in the paper. Thank you for these positive and constructive comments and suggestions.

We would like to express our great appreciation to you for comments on our paper. Looking forward to hearing from you.

  Thank you and best regards.

  Yours sincerely,

Corresponding author: Minyi Tian and Ying Zhou

mytian@gzu.edu.cn (M.T.); yingzhou71@yeah.net (Y.Z.)

Reviewer 2 Report

Although the authors present a well-organized and interesting manuscript, it for sure can benefit from several improvements based on the following recommendations and suggestions:

L12-13 rephrase to avoid repetition

Use keywords that are different from the title

Go throughout whole manuscript  for minor mistakes like in L92 where an inappropriate preposition has been used

The retention index can be added to Table 1 and Fig 1 can actually be omitted as non relevant to the reader. Compounds from Table 1 can be structured in chemical classes for easier interpretation.

Discussion in sections 2.1 and 2.2 is rather scarce.

Author Response

Chemical Composition, Antioxidant, Antibacterial, Enzyme Inhibitory, and Anticancer Activities of Alpinia galanga Flower Essential Oil

(Manuscript ID: pharmaceuticals-1886802)

Dear Reviewer:

  Thanks for your comments concerning our manuscript entitled “Chemical Composition, Antioxidant, Antibacterial, Enzyme Inhibitory, and Anticancer Activities of Alpinia galanga Flower Essential Oil” (Manuscript ID: pharmaceuticals-1886802). Those comments are all valuable and very helpful for revising and improving our paper, as well as the important guiding significance to our researches. We have studied comments carefully and have made correction which we hope meet with approval. Revised portion are marked in red in the paper. The main corrections in the paper and the responds to the reviewer’s comments are as flowing:

Responds to the reviewer’s comments:

For the second reviewer:

Reviewer 2:

Suggestion 1: “L12-13 rephrase to avoid repetition

  Response: In L14-15, “Alpinia galanga is widely cultivated as an important source of cosmetics, medicines, and culinary products, and its essential oil (EO) has been used in cosmetics and perfumes.” has been replaced by “Alpinia galangal is widely cultivated for its essential oil (EO), which has been used in cosmetics and perfumes.”. Revised portions are marked in red in the paper. Thank you for these positive and constructive comments and suggestions.

Suggestion 2: Use keywords that are different from the title

Response: In keywords, “Alpinia galanga; essential oil; chemical composition; antioxidant activity; antibacterial activity; enzyme inhibitory activity; anticancer activity” has been replaced by “Alpinia galanga flower; farnesene; radical scavenging effects; antibacterial agent; enzyme inhibitors; cytotoxicity; apoptosis”. Revised portions are marked in red in the paper. Thank you for these positive and constructive comments and suggestions.

Suggestion 3: Go throughout whole manuscript for minor mistakes like in L92 where an inappropriate preposition has been used

Response: In L95, “Compared with the above study,” has been replaced by “In contrast to the above-mentioned studies,”. In addition, we have carefully checked the entire manuscript, and revised portions are marked in red. Thank you for these positive and constructive comments and suggestions.

Suggestion 4: The retention index can be added to Table 1 and Fig 1 can actually be omitted as non relevant to the reader. Compounds from Table 1 can be structured in chemical classes for easier interpretation.

Response: In the table 1, we have added the percentages of monoterpene hydrocarbons, oxygenated monoterpenes, sesquiterpene hydrocarbons, oxygenated sesquiterpenes, and others. Revised portions are marked in red in the paper. Thank you for these positive and constructive comments and suggestions.

Suggestion 5: Discussion in sections 2.1 and 2.2 is rather scarce.

Response: In sections 2.1 and 2.2, we have added related discussions. In section 2.1, “In addition, the major volatile compounds of the hexane extract from A. galanga flower (obtained from Gainesville, USA) were α-humulane, pentadecane, β-farnesol, dimethyl trisulfide, and mercaptomethylbutanol [8].” has been added. In section 2.2, “Aceteugenol, an acetylated derivative of eugenol, had significant activity in scavenging DPPH free radicals with IC50 of 0.12 ± 0.03 μmol/L, and it showed a synergistic effect with eugenol on inhibiting the oxidation of sunflower oil [42]. Besides, isoeugenol is an isomer of eugenol, and methyleugenol is a methylated derivative of eugenol. These eugenol-related compounds may play a key role in the antioxidant activity of A. galanga flower EO.” has been added. Revised portions are marked in red in the paper. Thank you for these positive and constructive comments and suggestions.

We would like to express our great appreciation to you for comments on our paper. Looking forward to hearing from you.

  Thank you and best regards.

  Yours sincerely,

Corresponding author: Minyi Tian and Ying Zhou

mytian@gzu.edu.cn (M.T.); yingzhou71@yeah.net (Y.Z.)
